# Three heads are better than two: Comparing learning properties and performances across individuals, dyads, and triads through a computational approach

Tsutomu Harada *

Graduate School of Business Administration, Kobe University, Kobe, Japan

* harada@people.kobe-u.ac.jp

**Data Availability Statement:** The data are available at OSF: https://osf.io/sa9n2/.

**Funding:** This work was supported by JSPS KAKENHI under Grant (Number 26380506).

## Abstract

Although it is considered that two heads are better than one, related studies argued that groups rarely outperform their best members. This study examined not only whether two heads are better than one but also whether three heads are better than two or one in the context of two-armed bandit problems where learning plays an instrumental role in achieving high performance. This research revealed that a U-shaped correlation exists between performance and group size. The performance was highest for either individuals or triads, but the lowest for dyads. Moreover, this study estimated learning properties and determined that high inverse temperature (exploitation) accounted for high performance. In particular, it was shown that group effects regarding the inverse temperatures in dyads did not generate higher values to surpass the averages of their two group members. In contrast, triads gave rise to higher values of the inverse temperatures than their averages of their individual group members. These results were consistent with our proposed hypothesis that learning coherence is likely to emerge in individuals and triads, but not in dyads, which in turn leads to higher performance. This hypothesis is based on the classical argument by Simmel stating that while dyads are likely to involve more emotion and generate greater variability, triads are the smallest structure which tends to constrain emotions, reduce individuality, and generate behavioral convergences or uniformity because of the "two against one" social pressures. As a result, three heads or one head were better than two in our study.

## Introduction

Many people believe in the promise of teamwork and synergy, necessitating that the whole is greater than the sum of the parts; "two heads are better than one" or "none of us is as smart as all of us." However, the number of heads yielding optimal levels of synergy in decision-making remains unknown. While synergies could be generated by knowledge sharing or diverse perspectives, it is difficult to accurately specify the exact factors associated with synergy. Hence, this study addressed this question by taking a computational approach to group decision-making in a simple Q learning model with two options and statistically identified the causes of synergy.

**Competing interests:** There are no conflicts of interest to declare.

Notably, some of the related studies emphasized the significance of team synergy. Surowiecki [1] denoted through numerous case studies that the collective wisdom of a large group of individuals proves to be correct more often than the judgment of a single decision-maker. Nevertheless, this result hinges on the conditions that (1) each person should have private information even if it happens to be an eccentric interpretation of known facts; (2) people's opinions are not determined by the opinions of those around them; (3) people can specialize and draw on local knowledge; (4) some mechanisms exist for turning private judgments into a collective decision; (5) each person trusts the group as a whole to be fair. If these conditions are violated, a number of dysfunctional dynamics occur, resulting in ineffective group pressure or groupthink.

Extensive work by past studies on collective decision-making has reaffirmed the proposition that groups rarely outperform their best members [2, 3]. One robust evidence for the determinants of group performance is the fact that group decisions are governed by a confidence heuristic [4, 5]. This notion implies that group discussions are dominated by the more confident members in a group, and their responses are more likely to be correct than those of less confident members in general [6]. Recently, the ability of groups to combine individual information has been intensively examined through signal detection experiments concerning group decision-making [6–14]. Barami et al. [9] concurred that interactive decision-making between two individuals was better than one when they shared a similar visual sensitivity, and when presented with an equal opportunity to communicate freely. Nevertheless, if two individuals exhibit different visual sensitivities, their performance is generally worse than that of one head. These findings were accounted for by the weighted confidence sharing model in which two heads accurately communicate their level of confidence on every trial. In another study, Bahrami et al. [8] revealed that groups wherein members were heterogeneous in terms of perceptive abilities tend to perform poorly. The lower performance of heterogeneous groups implies that the way the individual information is aggregated is not necessarily efficient. According to Bahrami et al. [9],' this is because groups use a suboptimal decision rule that puts more weight on the information provided by the least able member. Thus, when a greater difference exists between group members' information reliabilities, the resulting efficiency loss increases.

Apart from signal detection experiments, Woolley et al. [15] distinguished collective intelligence that did not strongly correlate with the average or maximum individual intelligence of group members but corresponded well with the average social sensitivity of group members, the equality in distribution of conversational turn-taking, and the proportion of females in the group. Several investigations regarding organizational behavior necessitate the negative aspects of teams, such as group pressure [16–21], risky shift [22, 23], social loafing [24], interpersonal competition [2], and group thinking [25, 26], leading to group collective unintelligence.

While the signal detection approach succeeded in formulating rigorously the underlying group dynamics in decision-making, it should be noted that group decision-making in such experiments did not involve intertemporal learning. Instead, they highlighted simple perception problems. Yet, in more complicated problems which do not assume correct solutions, the internal mechanism of sharing information and confidence across interacting members in the WCS does not necessarily lead to higher performance. Instead, sharing consistent learning rules seem more important. Moreover, while the signal detection approach highlighted the problem of individual vs. dyad, the effects of more group size on performance remain to be examined. In particular, this study is interested in examining whether three heads perform better than four or not.

Two heads vs. three problem highlights new interesting issues in group decision-making which do not arise in the two heads vs. one, i.e., even-sized groups vs. odd-sized groups [2, 27]. Small groups are likely to break into two coalitions. If a group is even-sized, two subgroups are equal in size. In this case, since the majority rule cannot be applied, subgroup dynamics might

lead to deadlock [28–31]. In contrast, if a small group is odd-sized, a minority and a majority subgroups emerge, and the majority influence provides a clear direction and group cohesion [2, 27, 32, 33].

In our context, this argument could be related to the coherence of group learning. Because majority rule cannot be applied to dyads, decision making in learning situations may eventually become incoherent. In one moment, one member may make a decision based on her learning preference, and in another moment, another member may take initiative in decision making. As a result, group learning strategy is likely to become incoherent over time. In contrast, because majority rule can be applied to triads, majority subgroups may make decisions based on their own learning strategies; thus, group learning strategies may be more coherent. Consequently, triad learning performance may outperform dyad learning performance due to the former's learning coherence and the latter's learning incoherence. Notably, both triads and individuals can pursue coherent learning strategies; hence, relative performance may not be predicted in advance without imposing further conditions. However, we may predict that an inverted U-shaped relationship in learning performance emerges across indivisuals, dyads, and triads. S1 Appendix presents a simple model of individual and triad learning coherence and dyad learning incoherence.

Thus, this study's main hypothesis was that a U-shaped relationship emerges across individuals, dyads, and triads because learning coherence is more likely in individuals and triads and learning incoherence is more likely in dyads. To test this hypothesis, this study arranged several experimental settings for small groups who predominantly relied on online face-to-face communication. Most of the members were not acquaintances and were communicating with each other for the first time, which controlled for the effects of group pressures and secured psychological safety. This stratagem was viable because participants did not have to worry about any personal relationships, and focused their attention on group tasks. Under this controlled environment, this study attempted to identify how group decision-making differs from that of individuals in terms of performance and learning properties such as the exploitation/ exploration ratio. This study ran experiments conducted by individuals, dyads, and triads so that the effects of group size from one to three could be evaluated. Hence, this study could test the efficacy of both two heads and three heads in comparison to one.

Furthermore, this study did not depend on signal detection experiments as this study was more interested in the learning properties of group decision-making in contrast to information sharing and filtration. Thus, this study adopted a reinforcement learning (RL) framework [34] to account for decision-making and learning behaviors in the two-armed bandit (TAB) problems, which is the standard model for model-based analysis of choice behavior. The RL framework has been extensively studied in the context of multi-armed bandit problems, in particular, closely associated with neural signals in cortical and subcortical structures [35–38]. Moreover, the RL framework has also been adopted to study learning behavior in many social contexts [39–45]. Nevertheless, to the best of our knowledge, this framework has not been applied to the study of group decision-making. One advantage of taking this computational approach is that learning parameters could be estimated as groups and also compared across and within groups of different sizes. This new approach to group dynamics allowed us to rigorously estimate and characterize the properties of group dynamics.

## Methods

### Participants

The experiment in this study was implemented in one of the undergraduate courses the author taught at Kobe University. Initially, a total of 336 students participated in the experiment for

course credit, but 14 participants were excluded before the analysis because they took only one of the three tests in this experiment. As a result, the sample in this study consisted of a total of 322 healthy undergraduate students (i.e., 100 females, age range = 19–25 years, SD = 1.21). All participants and their academic advisers signed informed consent before the experiment, which was approved by the local Ethics Committee at the Graduate School of Business Administration, Kobe University.

## Experiment

In Test 1, each participant undertook the TAB independently. In Test 2, participants formed a pair and undertook the TAB. In Test 3, participants played the TAB in groups of three. In total, this study conducted seven sessions of the TABs using the online communication software, Zoom. In each session, three tests were randomly assigned to participants using the breakout sessions in Zoom to control for learning effects. All tests were performed with PsytoolKit [46, 47]. Group members in Tests 2 and 3 freely communicated via Zoom during the session, while sharing test screens in PsytoolKit, and made choices. Participants were required to complete the tests within 40 minutes. Most of the groups finished the tests within 30 minutes. Additionally, there was a one-week interval between the two successive sessions.

All participants in this sample undertook Test 1 once and at least one more experiment in Tests 2 or 3. In Test 2 (dyads), 230 and 23 participants played the TAB once and twice, respectively. In Test 3 (triads), 153, 66 and 13 participants played one, two, and three times, respectively. Since Test 3 required more participants than Test 2, the number of those who participated in the entire experiment, more than once, were more than those who participated in Test 2. Random assignment of participants mitigated their learning effects across Tests 2 and 3. For example, participants might have experienced two rounds of Test 3 first, and then, participated in Test 2. In this case, their learning might be carried over to the last experiment. By randomizing the order of assignment to experiments, these effects were expected to be mitigated in the pooled sample. The total numbers of groups of dyads and triads were 138 and 108, respectively.

For the purpose of group comparison, we also constructed a subsample in which at least one member of the dyadic and triadic groups undertook Test 3 and Test 2, respectively. In total, 262 individuals were in this subsample. In this subsample, 116 dyadic groups and 93 triadic groups were identified. In Tests 2 and 3, the numbers of individuals who took the tests one, two, three, and four times were 84, 118, 49, and 11, respectively. Of the dyads, 216 individuals participated, and 200 and 16 undertook Test 2 once and twice, respectively. Of the triads, 205 individuals participated, and 142, 52 and 11 individuals undertook Test 3 one, two and three times, respectively.

## Two-armed bandit problem

In the TAB problem, participants complete a series of 100 choices from two boxes. In each choice, participants selected either a right or left box, and immediately after clicking, the reward appeared (Fig 1). The reward was either 10 points or 0 for a choice and participants are required to maximize the total rewards out of the 100 choices.

One of the boxes was advantageous with a 70% success probability and the other was disadvantageous with a 30% success probability. Alternatively, after a certain number of trials, advantageous and disadvantageous cards switched to disadvantageous and advantageous ones, respectively. During the 100 trials, these switches were designed to intervene three times, the timing of which varied across experiments. For instance, in a few experiments, the switches took place on the 30$^{th}$ and 70$^{th}$ trials such that the right card was advantageous for the first 30

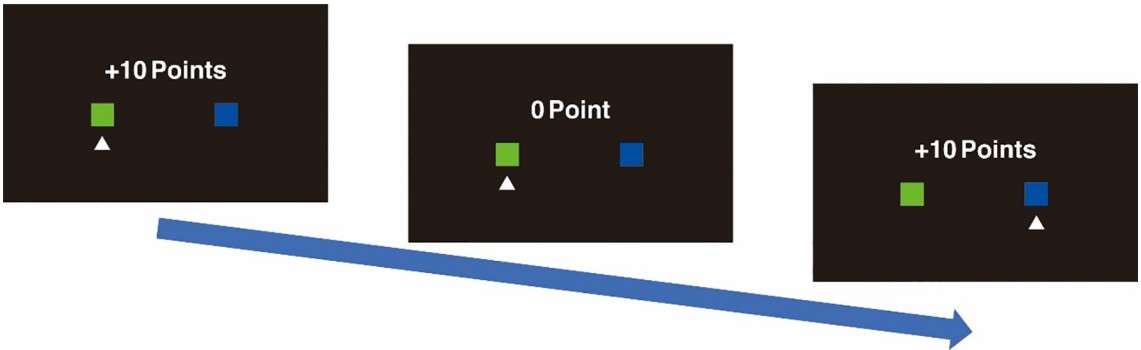

**Fig 1. Two-armed bandit problems (TAB).** Participants chose and clicked either on the right (blue) or left (green) box. Immediately after clicking, the reward of either +10 points or 0 points appears. In this figure, the left box is selected first with 10 points, followed by the left and right boxes with the rewards of 0 points and 10 points, respectively. Participants undertake this selection 100 times to maximize total rewards.

trials, but became disadvantageous from 31st to 70th trials, and reverted to being advantageous in the last 30 trials. The timings and success probabilities were not known to participants. These settings were designed because learning convergence was likely to be achieved in the first 30–60 trials in our past experiments. Once the convergence was achieved, participants only selected the same box afterwards, which in turn biased the estimates of learning parameters.

**Q learning model.** This study considered two types of Q learning models [48] to estimate learning parameters in the TAB. First, in the simple Q learning model, the action value $Q_i(t)$ of the chosen option i at trial t is updated as:

$$Q_i(t+1) = Q_i(t) + \alpha\delta(t), \tag{1}$$

with

$$\delta(t) = R_i(t) - Q_i(t), \tag{2}$$

where $R_i(t)$ and $\alpha(0<\alpha\leq 1)$ are the reward of the option i at trial t and the learning rate, respectively. $\delta(t)$ refers to the reward prediction error, measured by the difference between the current value estimate and the obtained reward R. The action value reflects immediate reward by scaling the prediction error with the learning rate. If learning rates are close to 1, fast adaptations are made based on prediction errors. If learning rates are closer to 0, adaptation becomes very slow. The initial action values are set to zero so that $Q_i(1) = 0$ for i = 1,2.

For the unchosen option j (i≠j), the action value remains the same as before:

$$Q_j(t+1) = Q_j(t). \tag{3}$$

Denote the chosen action at trial t by a(t)∈{1, 2}. The probability to choose either option is specified via the following softmax decision rule:

$$P(a(t) = i) = \frac{exp(\beta Q_i(t))}{\sum_{j=1}^{4} exp(\beta Q_j(t))}, \tag{4}$$

where $P(a(t) = i)$ indicates the probability to choose the action $a(t) = i$ at trial t. The parameter β refers to the inverse temperature, a parameter that assesses the relative strength of exploitation vs. exploration. Exploitation indicates "the optimization of current tasks under existing information and memory conditions", while exploration signifies "wider and sometimes

random searches and trials that do not coincide with the optimal solutions provided by the exploitation" [49]. A higher β value suggests that the choices are to be made primarily based on the action value Q, implying exploitation. Contrarily, a lower β value indicates more random choices, regardless of the action value Q, implying exploration. This is because the relative importance of the Q value in Eq (4) declines significantly. Hence, the inverse temperature β refers to the relative weight of exploitation against exploration in decision-making.

**Asymmetric Q learning model.** This Q learning model assumes that learning parameter α was assumed to be symmetric, regardless of the sign of the reward prediction error δ(t). However, related studies showed that the learning rates were asymmetric [50–55]. Thus, the asymmetric Q learning model was considered, incorporating asymmetric learning parameters. In this model, the action value $Q_i(t)$ of the chosen option i is updated via the following rule:

$$Q_i(t+1) = \begin{cases} Q_i(t) + \alpha^+ \delta(t) + \phi \ \textit{if} \ \delta(t) \geq 0, \\ Q_i(t) + \alpha^- \delta(t) + \phi \ \textit{if} \ \delta(t) < 0, \end{cases} \tag{5}$$

where $\alpha^+$ and $\alpha^-$ are the learning rates when the reward prediction errors are positive (or zero) or negative, respectively. The idea behind this specification is the positivity biases. Cazé and van der Meer [56] showed that even in simple, static bandit tasks, agents with differential learning rates can outperform unbiased agents. They suggested the existence of a situation in which the steady-state behavior of asymmetric RL models yields better separation of the action values compared with symmetric RL models [56]. While this proposition was proved mathematically as an asymptotic property, real performance in cognitive tasks includes not only asymptotic properties but also transient outcomes [57]. $\phi$ is added here as the choice trace to account for autocorrelation of choice, which could affect the learning biases [57]. For the unchosen option j (i≠j), the action value is updated according to Eq (3), and the probability to choose the option is computed via the softmax decision rule in Eq (4).

## Estimation method

The parameters clarified in the model were estimated by optimizing the maximum a posteriori (MAP) objective function, that is, finding the posterior mode:

$$\hat{\theta} = argmax \ p(D_s|\theta_s)p(\theta_s), \tag{6}$$

where $p(D_s|\theta_s)$ is the likelihood of data $D_s$ for subject s, and conditional on parameters $\theta_s = \{\alpha^S, \phi^S, \beta^S\}$ for the simple Q learning model and $\theta_s = \{\alpha^{\pm S}, \phi^S, \beta^S\}$ for its asymmetric version. $p(\theta_s)$ is the prior probability of $\theta_s$. This study assumed each parameter is bounded and uses constrained optimization to find the MAP estimates. More specifically, since α ($\alpha^\pm$) is bounded between 0 and 1, and β, take non-negative values, their priors were assumed to follow beta distributions for α ($\alpha^\pm$), and gamma distributions for β.

## Results

To reveal the group effects, group differences across individuals, dyads, and triads and effects were examined, with respect to performance and learning parameters of the two Q learning models. After these analyses, the performance determinants were evaluated by regression analysis. The descriptive statistics for relevant variables were reported in Tables 1 and 2.

### Group differences

**Performance.** For the performances of the TAB, the number of gaining 10 points throughout 100 trials was used because the subsequent analysis applied the Poission regression

**Table 1. Descriptive statistics (pooled sample).**

|  | Individuals | | Dyads | | Triads | |
|---|---|---|---|---|---|---|
|  | **Mean** | **SD** | **Mean** | **SD** | **Mean** | **SD** |
| **Performance** | **49.57** | **5.28** | **48.41** | **5.24** | **50.36** | **7.06** |
| max | - | - | 52.37 | 4.22 | 53.95 | 3.99 |
| min | - | - | 46.71 | 4.53 | 45.54 | 4.19 |
| average | - | - | 49.54 | 3.75 | 49.86 | 3.26 |
| **(simple model)** | | | | | | |
| $\alpha$ | 0.57 | 0.25 | 0.59 | 0.26 | 0.6 | 0.24 |
| max | - | - | 0.7 | 0.19 | 0.79 | 0.14 |
| min | - | - | 0.41 | 0.24 | 0.33 | 0.18 |
| average | - | - | 0.55 | 0.19 | 0.56 | 0.14 |
| $\beta$ | 4.35 | 3.39 | 5.18 | 4.2 | 6.61 | 4.97 |
| max | - | - | 6.09 | 3.62 | 7.36 | 3.4 |
| min | - | - | 2.77 | 2.37 | 1.74 | 1.29 |
| average | - | - | 4.43 | 2.7 | 4.3 | 1.88 |
| **(asymmetric model)** | | | | | | |
| $\alpha^+-\alpha^-$ | 0.08 | 0.35 | 0.04 | 0.36 | 0.11 | 0.35 |
| max | - | - | 0.28 | 0.28 | 0.38 | 0.2 |
| min | - | - | -0.11 | 0.3 | -0.21 | 0.28 |
| average | - | - | 0.09 | 0.25 | 0.09 | 0.2 |
| $\beta$ | 4.35 | 3.02 | 4.81 | 3.34 | 5.77 | 3.71 |
| max | - | - | 6.08 | 3.23 | 6.69 | 2.59 |
| min | - | - | 2.78 | 2.19 | 1.95 | 1.42 |
| average | - | - | 4.43 | 2.39 | 4.18 | 1.59 |

N = 568.

dealing with counting data. The total sum of rewards is obtained if this performance measure is multiplied by 10. The average performance for individuals, dyads, and triads was respectively 49.57, 48.41, and 50.36. The Kruskal-Wallis test revealed the significant group size effects on performance ($\chi^2_2$ = 7.21, p = .03). Then, the pairwise Wilcoxon Rank-Sum Test with Bonferroni adjustment presented significant differences in performance existing between individuals and dyads (p = .06) and between dyads and triads (p = .05). Thus, the performance was higher for individuals and triads, and lowest for dyads, which suggested that three heads are better than two and one head is better than two.

**Table 2. Descriptive statistics (subsample).**

|  | Individuals | | Dyads | | Triads | |
|---|---|---|---|---|---|---|
|  | **Mean** | **SD** | **Mean** | **SD** | **Mean** | **SD** |
| **Performance** | **49.86** | **5.3** | **48.12** | **5.07** | **50.41** | **7.39** |
| **(simple model)** | | | | | | |
| $\alpha$ | 0.57 | 0.25 | 0.58 | 0.26 | 0.58 | 0.24 |
| $\beta$ | 4.4 | 3.41 | 5.18 | 4.32 | 6.59 | 5.03 |
| **(asymmetric model)** | | | | | | |
| $\alpha^+-\alpha^-$ | 0.1 | 0.35 | 0.03 | 0.36 | 0.11 | 0.35 |
| $\beta$ | 4.32 | 2.94 | 4.78 | 3.45 | 5.51 | 3.63 |

N = 471.

However, the data included groups whose members did not undertake other tests. For example, in some dyadic groups, some members did not complete Test 3 and in some triadic groups, some members did not complete Test 2. If these samples had been included, group comparisons between dyads and triads might have been imprecise; in other diadic and triadic groups, some group members completed both Tests 2 and 3. Hence, a similar analysis was conducted in the subsample of dyadic and triadic groups in which at least one group member undertook all tests.

The average performance for individuals, dyads, and triads was respectively 49.86, 48.12, and 50.41. The Kruskal-Wallis test showed the significant group size effects on performance ($\chi_2^2$ = 11.45, p = .003). The pairwise Wilcoxon Rank-Sum Test with Bonferroni adjustment presented significant differences in performance existing between individuals and dyads (p = .003) and between dyads and triads (p = 0.03). Thus, the results became more significant, showing a U-shaped relationship across individuals, dyads, and triads in this subsample.

If one head is better than two heads, it should naturally follow that two heads are better than three. However, in our experiments, three heads proved superior to two. This result was opposite to that presented by Bahrami et al. [9] who found two heads being better than one. This contrast could be accounted for by the differences in underlying cognitive tasks between the two studies. In Bahrami et al. [9], careful detection of oddballs was required whereas in our study, decision-making, given past performances, was given prime significance. In other words, the former cognitive task hinges on attention, but the latter requires further information processing.

**Learning parameters.** Consequently, the question arises that how were these results (i.e., three heads better than others and one head better than two) generated in group dynamics in Q learning tasks. The group dynamics, including group pressure and risky shifts, are generated as participants share experiences by working together. Gradually, group members begin to mutually exercise personal influences; however, in our study, such group dynamics could not take place by controlling amounts of communication across participants. Instead, group dynamics matter in this study with respect to learning related to the TAB. To inspect this mechanism, learning rates α, positivity biases $\alpha^+ - \alpha^-$, and the inverse temperature (exploitation/exploration ratio) β were estimated and compared across individuals, dyads, and triads.

The pooled sample was considered first. In the simple Q learning model, the Kruskal-Wallis test showed that no group difference exists for α ($\chi_2^2$ = 1.52, p = .47), but the inverse temperature generated group differences ($\chi_2^2$ = 15.09, p = 5.3e-04). The pairwise Wilcoxon rank-sum test with Bonferroni adjustment presented significant differences existing between individuals and triads (p = 3.3e-04) and between dyads and triads (p = .05).

In the asymmetric Q learning model, the positivity biases $\alpha^+ - \alpha^-$ were confirmed in individuals ($\chi_2^2$ = 20.42, p = 6.2e-06) and triads ($\chi_2^2$ = 12.53, p = 3.0e-04), but no posivitity biases were found in dyads ($\chi_2^2$ = 2.18, p = .14). In addition, $\beta$ showed the group size effects ($\chi_2^2$ = 6.98, p = .03). The pairwise Wilcoxon rank-sum test with Bonferroni adjustment presented significant differences between individuals and triads (p = .02).

Next, the subsample of groups in which at least one member completed all three tests was considered. In the simple Q learning model, the Kruskal-Wallis test indicated that no significant group differences existed for α ($\chi_2^2$ = .22, p = .90), but the inverse temperature generated group differences ($\chi_2^2$ = 10.95, p = .004). The pairwise Wilcoxon rank-sum test with Bonferroni adjustment presented significant differences between individuals and triads (p = .0003) and dyads and triads (p = .09). These results were quite similar to those in the pooled sample.

In the asymmetric Q learning model, the Kruskal-Wallis test confirmed positivity biases $\alpha^+ - \alpha^-$ in individuals ($\chi_2^2$ = 24.75, p = 6.5e-07) and triads ($\chi_2^2$ = 11.19, p = 8.2e-04), but not in dyads ($\chi_2^2$ = 1.29, p = .26). In addition, $\beta$ showed the group size effects ($\chi_2^2$ = 6.76, p = .03). The

pairwise Wilcoxon rank-sum test with Bonferroni adjustment presented significant differences between individuals and triads (p = .03). Once again, the results remained similar to those in the pooled sample.

## Within-group effects

Next, the effects for dyads and triads were examined. To see the group effects, the maximum, minimum, and the average of group members' individual performances and learning parameters were compared with the corresponding group variables. For example, the average of group members' individual performances was compared with the corresponding group performance. If the latter is higher, this suggests group effects are positive. The analysis was conducted with respect to the pooled sample because the comparison was made within each group, rather than across groups.

**Performance.** In dyads, the maximum and average of individual performances outperformed group performance ($\chi_1^2$ = 43.45, p = 4.4e-11 for the maximum and $\chi_1^2$ = 5.49, p = .02 for the average), but its minimum underperformed group performance ($\chi_1^2$ = 6.51, p = .01). Thus, group effects were positive in improving the minimum performance of group members, but it did not surpass the maximum and average performance of group members.

In triads, while the maximum of individual performances outperformed group performance ($\chi_1^2$ = 29.87, p = 4.6e-08), its minimum version underperformed group performance ($\chi_1^2$ = 36.47, p = 1.6e-09). However, the average of individual performances did not outperform group performance ($\chi_1^2$ = .06, p = .81). This suggests that in triads, group effects were higher in the sense that triads achieved higher performance than the minimum of individual performances and did not underperform compared to the average of individual performances.

**Learning parameters.** In dyads, the learning parameters of $\alpha$ in the simple model and the positivity biases $\alpha^+ - \alpha^-$ in the asymmetric model showed similar patterns in which the group parameters outperformed their minimums of individual group members ($\chi_1^2$ = 34.96, p = 3.4e-09 for $\alpha$, $\chi_1^2$ = 11.09, p = 8.7e-04 for $\alpha^+ - \alpha^-$) and underperformed their maximums ($\chi_1^2$ = 9.78, p = .002 for $\alpha$, $\chi_1^2$ = 29.74, p = 4.9e-08 for $\alpha^+ - \alpha^-$). However, while the group parameter of $\alpha$ outperfomed its average ($\chi_1^2$ = 4.08, p = .04), that of $\alpha^+ - \alpha^-$ was not statistically different from its average ($\chi_1^2$ = 1.56, p = .21).

Regarding the inverse temperature, $\beta$, the group parameters outperformed the minimums of individual group members ($\chi_1^2$ = 21.50, p = 3.5e-06 for the simple model, $\chi_1^2$ = 25.06, p = 5.6e-07 for the asymmetric model) and underperformed their maximum ($\chi_1^2$ = 7.00, p = .01 for the simple model, $\chi_1^2$ = 10.14, p = .001 for the asymmetric model); however, they did not differ from their averages ($\chi_1^2$ = .08, p = .78 for the simple model, $\chi_1^2$ = .17, p = .68 for the asymmetric model).

Thus, group effects were mostly identified because group parameters surpassed the minimums of the corresponding individual group members. However, the group effects were not high enough to outperform the maximums of group members. In dyads, the inverse temperatures, $\beta$, in both models and the positivity biases in the asymmetric model neither outperformed nor underperformed their averages. Only the learning parameter $\alpha$ outperfomed its average.

In triads, all parameters, except $\alpha^+ - \alpha^-$ in the asymmetric model, showed a similar pattern in which the group parameters outperformed their averages and the minimums of individual group members but underperformed their maximums ($\chi_1^2$ = 4.46, p = .03, $\chi_1^2$ = 60.26, p = 8.3e-15, $\chi_1^2$ = 40.65, p = 1.8e-10 for the average, minimum, and maximum of $\alpha$, $\chi_1^2$ = 5.19, p = .02, $\chi_1^2$ = 69.18, p = 2.2e-16, $\chi_1^2$ = 4.27, p = .04 for the average, minimum, and maximum of $\beta$ in the simple model, $\chi_1^2$ = 7.36, p = .01, $\chi_1^2$ = 65.24, p = 6.6e-16, $\chi_1^2$ = 5.32, p = .02 for the average, minimum, and maximum of $\beta$ in the asymmetric model). Meanwhile, the maximum and minimum

of $\alpha^+ - \alpha^-$ in the asymmetric model respectively outperformed and underperformed the group parameters ($\chi_1^2 = 31.61$, p = 1.9e-08, $\chi_1^2 = 43.08$, p = 5.2e-11 for the maximum and minimum); however, the average of $\alpha^+ - \alpha^-$ was not statistically different from the group parameters ($\chi_1^2 = .64$, p = 0.42).

Hence, both in dyads and triads, all group learning parameters outperformed the minimums and underperformed the maximums of group individuals. The difference emerged with respect to averages. In most cases, group parameters outperformed averages. However, the group parameters of the inverse temperatures in both simple and asymmetric models in dyads and the positivity biases in dyads and triads were not statistically different from their averages. Thus, differentiating factors in terms of within-group effects between dyads and triads were identified in the inverse temperature in both simple and asymmetric models; triads outperformed, but dyads did not surpass their averages.

## Determinants of performance

The results on performance suggest that a U-shaped relationship exists between group size and performance. Besides, β seemed to account for higher performance. To examine this more rigorously, this study regressed group dummy variables (Individuals, Dyads, Triads) and the learning parameters on performance. For this group comparison, the regression analysis was based on the subsample. The results in the simple and asymmetric models are respectively shown in Tables 3 and 4.

According to the tables, overall, Individuals and Triads positively, but Dyads negatively, accounted for performance. Moreover, while the coefficient of Size was negative, the Size squared was positive in both models, attesting the existence of the U-shaped relationship

**Table 3. Determinants of performance (simple Q learning model) (SE in parentheses).**

| Variables | Performance | | | | | | |
|---|---|---|---|---|---|---|---|
| | (1) | | (2) | | (3) | | |
| Constant Terms | 46.58 | *** | 48.46 | *** | 54.26 | *** | |
| | (1.06) | | (0.91) | | (2.57) | | |
| α | 1.13 | | 1.13 | | 1.13 | | |
| | (1.29) | | (1.29) | | (1.29) | | |
| β | 0.17 | ** | 0.17 | ** | 0.17 | ** | |
| | (0.08) | | (0.08) | | (0.08) | | |
| Individual | 1.88 | ** | | | | | |
| | (0.78) | | | | | | |
| Dyad | | | -1.88 | ** | | | |
| | | | (0.78) | | | | |
| Triad | 2.03 | ** | 0.15 | | | | |
| | (0.98) | | (0.87) | | | | |
| Size | | | | | -7.75 | *** | |
| | | | | | (3.01) | | |
| Size squared | | | | | 1.96 | ** | |
| | | | | | (0.77) | | |
| AIC | 3006.4 | | 3006.4 | | 3006.4 | | |

N = 471. The dependent variable is performance. Individual, Dyad, and Triad are dummy variables for individual, dyad, and triad. Size is the number of participants, which is 1, 2, and 3 for respectively individual, dyad, and triads. Since it takes only non-negative counting values, the Poison regression was applied to achieve statistical consistency.

** and *** Symbols indicate p < .05, and p < .01, respectively.

**Table 4. Determinants of performance (asymmetric Q learning model).** (SE in parentheses).

| Variables | Performance | | | | | |
|---|---|---|---|---|---|---|
| | (1) | | (2) | | (3) | |
| Constant Terms | 46.58 | *** | 48.50 | *** | 54.39 | *** |
| | (0.82) | | (0.62) | | (2.50) | |
| $\alpha^+ - \alpha^-$ | -0.70 | | -0.70 | | -0.70 | |
| | (0.94) | | (0.94) | | (0.94) | |
| β | 0.34 | *** | 0.34 | *** | 0.34 | *** |
| | (0.10) | | (0.10) | | (0.10) | |
| Φ | 0.01 | | 0.01 | | 0.01 | |
| | (0.02) | | (0.02) | | (0.02) | |
| Individual | 1.92 | ** | | | | |
| | (0.79) | | | | | |
| Dyad | | | -1.92 | ** | | |
| | | | (0.79) | | | |
| Triad | 2.05 | ** | 0.13 | | | |
| | (0.98) | | (0.87) | | | |
| Size | | | | | -7.87 | *** |
| | | | | | (3.03) | |
| Size squared | | | | | 1.99 | ** |
| | | | | | (0.78) | |
| AIC | 3002.5 | | 3002.5 | | 3002.5 | |

N = 471. The dependent variable is performance. Individual, Dyad, and Triad are dummy variables for individual, dyad, and triad. Size is the number of participants, which is 1, 2, and 3 for respectively individual, dyad, and triads. Since it takes only non-negative counting values, the Poison regression was applied to achieve statistical consistency.

** and *** Symbols indicate p < .05, and p < .01, respectively.

between group size and performance. Regarding learning parameters, the inverse temperature β in both models significantly accounted for performance. Thus, Individuals, Triads, and β were the determinants of higher performance in the TAB.

## Model fits

Finally, to compare the two models mentioned above, WBICs [58] was calculated for both models. The average WBICs were -49.11 for the simple model and -50.58 for the asymmetric model. The difference was tested and the result indicated no significant difference between the two (T(1134) = 1.34, p = .18), implying that the two models cannot be differentiated statistically in the pooled sample.

In each group, WBICs were also calculated. In individuals, the average WBICs of the simple and asymmetric models were -50.97 and -52.64, respectively, and no statistical difference was identified (T(642) = 1.19, p = .23). In dyads, the average WBICs of the simple and asymmetric models were -48.94 and -50.35, respectively, and once again, no statistical difference was identified (T(274) = .62, p = .54). Similarly, in triads, the average WBICs of the simple and asymmetric models were -43.75 and -44.72, respectively, and no statistical difference was identified (T(214) = .36, p = .71). Therefore, the two models cannot be statistically differentiated. Nevertheless, the fact that both models generated a high correlation between inverse temperatures and performance and similar patterns in within-group effects confirmed the robustness of the results with respect to model specifications.

## Discussion

One of the interesting findings in our study was that a relationship between performance and group size was validated to be U-shaped. As the regression analysis revealed, the causes for this performance difference could be attributed to higher values of the inverse temperatures $\beta$ in both models. In dyads, group effects regarding the inverse temperatures in both models did not generate higher values to surpass their averages, which might lead to lower performance. In contrast, triads gave rise to higher values of the inverse temperatures than their averages of group members. These differences are responsible for the U-shaped relationship in performance. Although the model selection tests did not differentiate between the simple and asymmetric Q learning models, both shared the same results that the inverse temperature $\beta$ accounted for higher performance. Thus, our results are robust to model specifications.

At individual levels, participants were more likely to perform the two-armed bandit game in an exploratory manner because their inverse temperatures were relatively lower to dyads and triads. The emphasis on exploration at individual levels indicate that rationality in terms of exploitation in the framework of the underlying learning model increased as more group members were added to the group decision-making processes. To achieve agreement in groups, logical reasoning and persuasion based on rational calculation would be required instead of exploration. Yet, in dyads, this increase in exploitation was not sufficient to make it significantly different from individuals. Indeed, group effects could not generate higher values of $\beta$ than its averages. It could be inferred that dyads encountered learning incoherence, leading to smaller group effects regarding the inverse temperature.

According to Simmel [59], in dyads, social interaction is more personal, involving more affect or emotion, and generates greater variability. The negative aspect of social interaction seemed to appear in dyads in our experiments. On the other hand, Simmel [59] argued that triads are the smallest structure that tends to constrain emotions, reduce individuality, and generate behavioral convergences or uniformity because of the "two against one" social pressures. These forces form the basis for uniformity, emergent norms, and cohesion [60]. Consequently, while dyads failed to improve the inverse temperature beyond its average as a result of affective or emotional influences, the smallest social structure, in the form of a triad, improved efficiency due to social pressures and more exploitation. This is also consistent with the theoretical hypothesis in S1 Appendix where dyads are likely to adopt more randomized learning strategies, whereas individuals and triads adopt coherent learning strategies. Although individuals might use more exploratory behaviors, exploration itself is one of the coherent learning strategies. Hence, our empirical results support our hypothesis that learning incoherence takes place in dyads but not in triads.

Notably, the positivity biases were confirmed for individuals and triads, but no such learning biases existed for dyads. As related studies indicated [50–55], learning biases are more likely in such leaerning situations. This result further evidences learning coherence in individuals and triads and learning incoherence in dyads.

Apart from this main result, the fact that group parameters achieved higher values than its means of individual members in most of the learning parameters deserves some attention in its own right. Not only triads, but also dyads, had these positive effects. Future studies should explore these group effects in more detail.

However, our findings are subject to several limitations. First, the results critically depend on the tasks that the groups perform and the learning situations where the TAB games are played. Different game settings could lead to different results. Second, learning properties could change over time through learning, therefore, their reliability might be subject to some limitations. Performance probably changed as participants undertook more TAB games,

because of the stochastic nature of the rewards. However, it could be conjectured that its learning strategy tends to be relatively stable because participants could not fully detect the stochastic environments (i.e., which options are more likely to generate higher rewards), as the probability of obtaining higher gains was changed twice during the 100 trials. Hence, it seems that participants were less likely to change their learning strategies even when they undertook the TAB several times. This justifies the use of learning properties in this study. Nevertheless, the reliability of learning properties should be tested in a future study.

Third, although this study used a relatively large sample, different results could be found in different samples, in particular, in different cultural contexts. For example, Shen et al. [61] noted that, when examining the effects of risk-taking on convergent thinking, they found that risk-taking was negatively associated with convergent thinking in China, but these correlations were close to zero or negative in the Netherlands. Thus, cultural effects could alter the learning strategies in the TAB, and hence, the effects of group dynamics on group performance.

Despite these limitations, the findings in this study deserve some attention because previous studies did not evaluate and examine the effects of group dynamics in terms of learning properties. Moreover, the results are intuitive and consistent with the simple hypothesis that the U-shaped relationship with respect to performance emerged due to the coherence of learning strategies. Even though these results might not be supported in different experimental settings; our computational approach could still be applied and is expected to generate new results. Thus, the contribution in this study would be more methodological. This study encourages future research that examines the learning mechanism of group dynamics, according to the computational approach suggested in this study.

## Supporting information

**S1 Appendix.**
(DOCX)

## Author Contributions

**Conceptualization:** Tsutomu Harada.

**Data curation:** Tsutomu Harada.

**Formal analysis:** Tsutomu Harada.

**Funding acquisition:** Tsutomu Harada.

**Investigation:** Tsutomu Harada.

**Methodology:** Tsutomu Harada.

**Project administration:** Tsutomu Harada.

**Resources:** Tsutomu Harada.

**Software:** Tsutomu Harada.

**Supervision:** Tsutomu Harada.

**Validation:** Tsutomu Harada.

**Visualization:** Tsutomu Harada.

**Writing – original draft:** Tsutomu Harada.

**Writing – review & editing:** Tsutomu Harada.

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
