## [Decision Letter · Decision Letter 0]

19 Jan 2021

PONE-D-20-35580

Three heads are better than two: Comparing learning properties and performances across individuals, dyads, and triads through a computational approach

PLOS ONE

Dear Dr. Harada,

Thank you for submitting your manuscript to PLOS ONE. After careful consideration, we feel that it has merit but does not fully meet PLOS ONE’s publication criteria as it currently stands. Therefore, we invite you to submit a revised version of the manuscript that addresses the points raised during the review process.

Two experts in the field reviewed your work.  I also read the paper myself, as I find this an intriguing topic, related to my own research.  The reviewers seemed to find the topic of your study interesting, and perhaps intriguing; however, they both note major issues with the paper that will need to be addressed, if this paper is to reach the bar for publication.  It seems a major issue is to improve the modeling work and data analysis to where it is consistent with similar work in the reinforcement learning and mathematical modeling literature.  You also need to better explain the key points and main goals for your study.  

We look forward to receiving your revised manuscript.

Kind regards,

Darrell A. Worthy, Ph.D

Academic Editor

PLOS ONE

Journal Requirements:

2.) Please change "female” or "male" to "woman” or "man" as appropriate, when used as a noun (see for instance https://apastyle.apa.org/style-grammar-guidelines/bias-free-language/gender).

3.) For this single-authored manuscript, please replace "we" with "I".

4.) We note that you have stated that you will provide repository information for your data at acceptance. Should your manuscript be accepted for publication, we will hold it until you provide the relevant accession numbers or DOIs necessary to access your data. If you wish to make changes to your Data Availability statement, please describe these changes in your cover letter and we will update your Data Availability statement to reflect the information you provide.

5.) Please include your tables as part of your main manuscript and remove the individual files. Please note that supplementary tables should  be uploaded as separate "supporting information" files.

6.) We noticed you have some minor occurrence of overlapping text with the following previous publication(s), which needs to be addressed:

- https://www.emerald.com/insight/content/doi/10.1108/S0065-2830(2010)0000032004/full/html

- https://www.sciencedirect.com/science/article/abs/pii/S0022249616301523?via%3Dihub

- http://www.wjh.harvard.edu/~cfc/Publications.html?

- https://www.sciencedirect.com/science/article/abs/pii/S0049089X13000884?via%3Dihub

In your revision ensure you cite all your sources (including your own works), and quote or rephrase any duplicated text outside the methods section. Further consideration is dependent on these concerns being addressed.

Reviewers' comments:

Reviewer's Responses to Questions

**Comments to the Author**

1. Is the manuscript technically sound, and do the data support the conclusions?

Reviewer #1: Partly

Reviewer #2: No

2. Has the statistical analysis been performed appropriately and rigorously? 

Reviewer #1: No

Reviewer #2: Yes

3. Have the authors made all data underlying the findings in their manuscript fully available?

Reviewer #1: No

Reviewer #2: No

4. Is the manuscript presented in an intelligible fashion and written in standard English?

Reviewer #1: Yes

Reviewer #2: Yes

5. Review Comments to the Author

Reviewer #1: Summary

The paper examines the question whether cooperative decision making can improve performance in 2-arm bandit task (TAB). Interactive decision making experiments are conducted online using ZOOM meeting software. A variant of Q-Learning is implemented to examine learning parameters (total reward, risk seeking, loss aversion, action selection noise) across the three conditions. Results are complex and do not present a straightforward interpretation by themselves. The discussion is, correspondingly, unclear and offers little clarity.

In the spirit of paper’s main research question, a colleague of mine and I reviewed the paper separately and exchanged comments to arrive at a joint review.

The paper has proposed a good question and used a remarkably inventive methodology to implement joint decision making in the time of pandemic. The paper could potentially make a valuable contribution. But there is a long way to go.

Major comments

There is no theory.

We do not know what to expect and WHY to expect it. Benefit in joint decision making is not the default (eg many works in joint Memory show that 2 people are worse than one). Could the author provide a formal model of dyadic and triadic performance that could produce some anticipated outcomes? In this regard, a very relevant paper to consider is Migdal et al (J Math. Psych. 2012).

The introduction discusses previous findings and theoretical claims regarding collective cognition and group intelligence. However, very little is written about reinforcement learning in general, and Q-learning in particular. Relevant issues include the reasons for choosing to focus on learning (as opposed to perceptual decision-making in previous research), the ecological validity of this task, and what we already know about Q-learning in individuals.

Moreover, the hypotheses and predictions should be fleshed-out and justified in the introduction. It is currently unclear what the hypotheses are. One can imagine that under a reasonable model of individual performance, it should be possible to simulate the dyadic and triadic decisions and have some predictions that could be directly compared to the data. At present, the only motivation to do the experiments seems to be to see “What could happen” and that can certainly be improved up on.

There is no model comparison.

We wondered to what extent the result depend on the exact implementation of Q-learning used in this study. It would be more convincing to show that the general pattern of results is consistent across a family of Q-learning models. For example, Equation 1 includes a constant noise that is not always present in Q-learning models. This equation also involves two learning rate parameters, one for negative and one for positive prediction errors. Do the results also hold in models with one learning rate, and without random noise? In a similar vein, do they also hold in a model that uses raw rewards, rather than prospect-utility-transformed values? Moreover, it can be illuminating to show potential differences in the best-fit model between the group sizes. For example, did a model with 2 learning rates fit the data better than a single-alpha model in all group sizes? This could be interesting, since it could be that not all group sizes weigh gains and losses differently (or equally).

The design is unclear

More clarity about design is needed. All subjects did the individual condition. Some did the dyadic (130 dyadic groups) and some did the triadic (110 triadic). These numbers show, and the paper indeed indicates that some participants did all three conditions but we do not know how many they were. Did any participants take part in more than 1 dyadic or triadic group? All of these issues make a big difference to establishing the right baseline.

For example, when we see the comparison of total collected reward in Figure 2, it makes more sense if the dyadic condition is compared to individual performance of the subjects who took part in dyadic condition and exclude subjects who did not. A similar issue applies to triadic condition which should have its own individual control.

The above would then allow the analysis of performance to be made not only between groups each group and the average of its individual but also between group and the best participant within a group.

Also, if possible, it would be advisable to try out comparing triads to the best dyad pairing within a group (see Wahn et al., 2018, PLoS one, for a similar approach studying visual search). These analyses will clarify whether the group-size differences reflect statistical aggregation or genuine group dynamics.

Such an analysis should not only be made on the level of overall performance, but also on the level of model parameters. In other words, understanding the relationship between individual-level and group-level model parameters can illuminate the dynamics of group learning. But since there is a discrepancy between the number of people who took the dyadic and triadic conditions, I am not sure how feasible this option is.

More minor points:

1. p. 2: “causes of synergy” -> the term causes is too strong. Correlates? Factors associated with?

2. p. 4: “To convert the group from…”: sentence unclear. Please elaborate.

3. The author refers to a single individual as a group. This is a strange decision. It is more natural to use the terms “individual” for N=1, and “group” for N>1. Accordingly

4. P. 6 and onward: It is better to regard the study as comprised of a single, multi-condition experiment, rather than 3 different experiments.

5. P.7: how were the number of trials per run (before the probability reversals) distributed? How were the reversal points determined?

6. Procedure: who was the participant that responded in each trial? For example, did the participants take turns, or was the decision determined by the first participant who answered?

7. P. 10: what were the specific parameters of the distributions that were used as priors for parameter estimation?

8. Given the impressive sample-size used in this study, it would be very valuable to provide the readers access to the raw data, as well as the analysis code.

9. P. 11 and onward: exact p-values should be given for all statistical tests (significant or not).

10. P. 12, top: the difference between individuals and dyads does not meet the standard .05 criterion. The same is true for p. 14, top. However, I do not think that the Bonferroni correction is needed when having only 2 comparisons, so that the uncorrected p-value can be used in the former case.

11. P. 13, top paragraph: mode details should be given regarding the parameters that did not vary across group sizes (i.e., descriptive and inferential statistics).

12. A section on parameter recovery is missing. It is important to show the precision in which the parameter values could be recovered using the number of trials and participants used in this study. This can strengthen the claims regarding group-size invariance in some of the parameters.

13. Figures 2-4 do not provide much more information than is already given in the text, and hence should be omitted.

14. The bar graphs used to show the data are quite outdated compared to what is acceptable and standard practice these days which includes superimposing the data points on top of the bars and/or showing the distributions using violin plots and similar tools

Reviewer #2: This is an interesting study with (to my knowledge) a novel finding. In this study people performed a two-armed bandit task. They did this first individually, and later in a separate session they performed the same task as in groups of two or three people. The results suggest that groups of two people did worse than either individuals or groups of three people.

This is a good experiment and the main behavioral result is interesting and appears to be novel. But, there are some serious issues with the paper. It is not ready for publication as it is. In particular many aspects of the modeling are unclear. Without more detail it is not clear whether the model is appropriate, or whether it accurately characterizes the data.

Equation 1, phi is never explained. What does it represent and what function does it play in the model? It is also listed on pg. 10 as one of the parameters, but nothing is said about its prior.

I am also confused about the learning rate alpha. I’m guessing that the plus/minus superscript means there are separate learning rates for when the prediction error is positive or negative, but this should be stated explicitly (or explained if it is doing something else).

Also, alpha has a t subscript which implies it is dependent on the trial somehow, but if so, it is never explained.

I think that equation 3 is supposed to specify that you use the top part if R(t) is greater than 0, and the bottom part if it is less than 0.

But, on that note, the experiment only ever has positive rewards, so the bottom half of the equation would never be used. This means that with the present design the parameter v is never used and serves no purpose. It also means that (because there are never losses) this study cannot assess risk attitudes, so all sections of the paper related to risk are not valid. This is a major issue for the interpretations of the paper. It also calls into question whether the model is fitting well, and therefore whether analyses of other parameters (like inverse temperature) are meaningful.

The priors should also be better specified. The type of distributions are noted, but the parameters of those distributions are not mentioned.

Overall, the model as it was applied does not appear to be appropriate, and a number of parts of it are not adequately explained. It is also a relatively complicated model for a somewhat simple task—which is not necessarily a problem, but some of the choices need to be justified--such as using different learning rates for positive and negative prediction errors. Results of the model fit also needs more detail (i.e., fit statistics) so that we can assess how well it is characterizing the data.

Other issues and typos:

The paper implies that triads do better than individuals, but the difference is not significant, so the authors need to be more careful about how the results are presented. I noticed this in the abstract, but it might say it elsewhere as well

pg. 8 – the text jumps into explaining the modeling in the section that explains the task. It should probably be its own section

top of pg. 4 'put' should be 'puts'

pg. 5 "single" maybe should be "signal"

pg. 13. first sentence of 2nd paragraph. 'the' should be 'a'

6. PLOS authors have the option to publish the peer review history of their article (what does this mean?). If published, this will include your full peer review and any attached files.

Reviewer #1: No

Reviewer #2: No

---

## [Author Response · Author response to Decision Letter 0]

3 Mar 2021

We greatly appreciate reviewers’ helpful comments and we believe we could significantly improve the quality of the paper. Here are how we revised the manuscript in response to the comments. The editor requested “It seems a major issue is to improve the modeling work and data analysis to where it is consistent with similar work in the reinforcement learning and mathematical modeling literature. You also need to better explain the key points and main goals for your study”. We believe by responding to reviewer’' requests and comments, these challenges were satisfied in the current revised manuscript. 

Main changes

First, the main revisions in the current manuscript are as follows:

- In response to the comment by Reviewer 1 that no theory was presented, we described a very simple model in Introduction that analyzed the performance differences across individuals, dyads, and triads, leading to the U-shaped relationship, which was caused by the coherence and incoherence of learning.

- In response to Reviewer 2’s comment on the appropriateness of the Q learning model and suggestions to simplify the model and Reviewer 1’s inquiry into whether the results change or not when only one learning rate is introduced, we considered two models: (1) simple Q learning model with only one learning rate; (2) asymmetric learning model that only allowed for changing learning rates to the sign of reward prediction errors. The results in both models were the same: inverse temperature accounted for performance.

- Following the advice by Reviewer 1, we examined within-group effects in which group learning parameters and the maximum, average and minimum of its individual group members’ learning parameters were compared. 

- Following the request by Reviewer 1, in analyzing group differences, we examined not only the pooled sample, but also the subsample in which all group members experienced both dyads and triads experiments. This subsample consists of 161 individuals, 56 dyads, and 42 triads. Although a few differences emerges between the pooled and subsample, main results remained the same.

Responses to Reviewer 1:

1. There is no theory. We do not know what to expect and WHY to expect it. … Moreover, the hypotheses and predictions should be fleshed-out and justified in the introduction. It is currently unclear what the hypotheses are.

We introduced a simple model of learning of individuals, dyads, and triads, leading to the U-shaped relationship of group performance in the Introduction. This model follows the arguments of Simmel and proposition on even-sized groups vs. odd-sized group differences. We interpret these as learning coherence of odd-sized (individuals and triads in our manuscript) and learning incoherence of even-sized (dyads). The hypothesis was the U-shaped relationship across individuals, dyads, and triads, and learning inefficiency in dyads, which we believe were shown by our analysis. 

2. However, very little is written about reinforcement learning in general, and Q-learning in particular. Relevant issues include the reasons for choosing to focus on learning (as opposed to perceptual decision-making in previous research), the ecological validity of this task, and what we already know about Q-learning in individuals.

 In Introduction, we added “Thus, this study adopted a reinforcement learning (RL) framework (34) to account for decision-making and learning behaviors in the two-armed bandit (TAB) problems, which is the standard model for model-based analysis of choice behavior. The RL framework has been extensively studied in the context of multi-armed bandit problems, in particular, with close association with neural signals in various cortical and subcortical structures that behaved as predicted (35-38). Moreover, the RL framework has also been adopted to study decision-making and learning in various social contexts (39-45). Nevertheless, to the best of our knowledge, this framework has not been applied to the study of group decision-making. One advantage of taking this computational approach is that learning parameters could be estimated as groups and also compared across and within groups of different sizes. “

3. There is no model comparison. … Do the results also hold in models with one learning rate, and without random noise? In a similar vein, do they also hold in a model that uses raw rewards, rather than prospect-utility-transformed values? Moreover, it can be illuminating to show potential differences in the best-fit model between the group sizes.

 In the previous model, we only considered only one variant of Q learning model. In this revised version, we considered (1) simple Q learning model with only one learning rate; (2) asymmetric learning model, and compared the model fit by calculating WBIC in the subsection “Model fits” in Results. The statistical test did not differentiate between the two models. 

4. The design is unclear. … some participants did all three conditions but we do not know how many they were. Did any participants take part in more than 1 dyadic or triadic group? … it makes more sense if the dyadic condition is compared to individual performance of the subjects who took part in dyadic condition and exclude subjects who did not. A similar issue applies to triadic condition which should have its own individual control.

 Regarding the questions, we described “In test 2 (dyads), 23 participants played the TAB twice. In test 3 (triads), 72 participants played twice, and 14 participants played three times.” in Methods. In group comparison, we also considered the data in subsample in which all participants participated in dyads and triads, and compared the performance and learning parameters across three groups. 

5. Also, if possible, it would be advisable to try out comparing triads to the best dyad pairing within a group. ...Such an analysis should not only be made on the level of overall performance, but also on the level of model parameters.

 Following this, we compared group performance and learning parameters and the max, min, and average of individual members’ performance and learning parameters. The results were described in the subsection Within group effects in Results. 

Minor points

“causes of synergy”　⇒ factors associated with synergy

“To convert the group from…”: sentence unclear. ⇒ Deleted the sentence.

It is better to regard the study as comprised of a single, multi-condition experiment, rather than 3 different experiments. ⇒We rephrased “test 1”, “test 2”, and “test 3”, instead of experiment 1, 2, and 3.

P.7: how were the number of trials per run (before the probability reversals) distributed? How were the reversal points determined?

 Although we misunderstood this question, it was described that in one run, the probability reversals took place in the 31th and 71th trials. For the first 30 trials, the probability remained the same. Then, the probability was reversed, and up to 70th trial, the probability remained the same. The second reversal took place at the 71th trial, and the subsequent trials retained the same probability. Please let us know if this explanation (and that in the article) is still ambiguous. 

Procedure: who was the participant that responded in each trial? For example, did the participants take turns, or was the decision determined by the first participant who answered?

 In each trial, group members discuss and decide jointly. They communicated via breakout sessions in the Zoom. We described “Group members in tests 2 and 3 freely communicated via Zoom during the session, while sharing test screens in PsytoolKit, and made choices.” in Experiments in Methods. 

what were the specific parameters of the distributions that were used as priors for parameter estimation?

 We described in Methods: “More specifically, since α (α^±) is bounded between 0 and 1, and β, take non-negative values, their priors were assumed to follow beta distributions for α (α^±), and gamma distributions for β.”

Given the impressive sample-size used in this study, it would be very valuable to provide the readers access to the raw data, as well as the analysis code.

 Yes, we will upload the raw dataset immediately after the acceptance. 

P. 11 and onward: exact p-values should be given for all statistical tests (significant or not). ⇒ P values were added in all test results in the text. 

P. 12, top: the difference between individuals and dyads does not meet the standard .05 criterion. The same is true for p. 14, top. However, I do not think that the Bonferroni correction is needed when having only 2 comparisons, so that the uncorrected p-value can be used in the former case.

 This is correct. However, since we were interested in establishing the U-shaped relationship across three groups, we would like to adhere to the multiple comparison framework.

P. 13, top paragraph: mode details should be given regarding the parameters that did not vary across group sizes (i.e., descriptive and inferential statistics).

 The descriptive statistics were added in the text in Tables 1 and 2.

A section on parameter recovery is missing. It is important to show the precision in which the parameter values could be recovered using the number of trials and participants used in this study. This can strengthen the claims regarding group-size invariance in some of the parameters.

 This is important suggestion, but I am not sure whether this parameter recovery is feasible or not, given individual parameter estimations were conducted in this study. My understanding is that the parameter recovery could be applied to a few estimated parameters such as hierarchical Bayesian parameter estimation in which the common parameters are estimated across different individuals. In this study, each individual or each group is assumed to have different parameters. So we estimated parameters for each individual and each group. The total number of parameters amounted to 570 (322 individuals, 138 dyads, and 110 triads). For each sample, 2 parameters in the simple Q model, in which case, the total number of parameters is 1140. So, we are not sure how the parameter recovery for these 1140 estimation results could be reported in the paper. 

Figures 2-4 do not provide much more information than is already given in the text, and hence should be omitted. 14. The bar graphs used to show the data are quite outdated compared to what is acceptable and standard practice these days which includes superimposing the data points on top of the bars and/or showing the distributions using violin plots and similar tools

 These figures were deleted.

Responses to Reviewer 2:

Equation 1, phi is never explained. What does it represent and what function does it play in the model? It is also listed on pg. 10 as one of the parameters, but nothing is said about its prior.

 ϕ is added here because Q learning tends to generate autocorrelation of the choices, which might bias the estimates of learning parameters, as demonstrated by Katahira (2018). In the text, we added “ϕ is added here as the choice trace to account for autocorrelation of choice, which could affect the learning biases (56).”

 The comments on the previous learning model were correct and we changed the models completely. The two models were much simpler than the previous one, and more standard, used by many related studies. So we hope this time Reviewer 2 did not have any serious concerns about the models. 

The paper implies that triads do better than individuals, but the difference is not significant, so the authors need to be more careful about how the results are presented. I noticed this in the abstract, but it might say it elsewhere as well

 This is correct and we removed the corresponding sentences. Also in the simple model in Introduction, we showed that Individuals outperform triads if p<.5, but underperform otherwise. Thus, the relationship between the two should and was indeterminate. 

pg. 8 – the text jumps into explaining the modeling in the section that explains the task. It should probably be its own section

 We created the subsection “Two-armed bandit problem” and explained the game. Then, we moved on to the explanation of the models. 

top of pg. 4 'put' should be 'puts'

pg. 5 "single" maybe should be "signal"

pg. 13. first sentence of 2nd paragraph. 'the' should be 'a'

 We corrected all of these.

---

## [Decision Letter · Decision Letter 1]

7 Apr 2021

PONE-D-20-35580R1

Three heads are better than two: Comparing learning properties and performances across individuals, dyads, and triads through a computational approach

PLOS ONE

Dear Dr. Harada,

Thank you for submitting your manuscript to PLOS ONE. After careful consideration, we feel that it has merit but does not fully meet PLOS ONE’s publication criteria as it currently stands. Therefore, we invite you to submit a revised version of the manuscript that addresses the points raised during the review process.

I sent your paper back to the two original reviewers; R2 was satisfied with the revisions you made, but R1 still noted some concerns.  It seems the concerns center on the lack of attention to detail, as well as emphasizing the novel theoretical advances made by your paper.  I invite you to submit a revision, but please pay special attention to the points raised by R1.  An overarching concern is that this paper feels as though it was written in a hasty manner, simply to get another publication, and it needs to reach a certain level of quality before it is published.  Please be candid about noting the strengths and limitations of your study, so that the conclusions are supported by the data.  If you choose to submit a revision, I will evaluate the manuscript and decide whether to ask R1 to review it once again. 

We look forward to receiving your revised manuscript.

Kind regards,

Darrell A. Worthy, Ph.D

Academic Editor

PLOS ONE

Reviewers' comments:

Reviewer's Responses to Questions

**Comments to the Author**

1. If the authors have adequately addressed your comments raised in a previous round of review and you feel that this manuscript is now acceptable for publication, you may indicate that here to bypass the “Comments to the Author” section, enter your conflict of interest statement in the “Confidential to Editor” section, and submit your "Accept" recommendation.

Reviewer #1: All comments have been addressed

Reviewer #2: All comments have been addressed

2. Is the manuscript technically sound, and do the data support the conclusions?

Reviewer #1: No

Reviewer #2: Yes

3. Has the statistical analysis been performed appropriately and rigorously? 

Reviewer #1: No

Reviewer #2: Yes

4. Have the authors made all data underlying the findings in their manuscript fully available?

Reviewer #1: Yes

Reviewer #2: Yes

5. Is the manuscript presented in an intelligible fashion and written in standard English?

Reviewer #1: Yes

Reviewer #2: Yes

6. Review Comments to the Author

Reviewer #1: The paper has been substantially changed. But improved, I am not sure.

1. In the introduction, we see an attempt at motivating a theoretical justification in pp. 6-7. Frankly, I did not understand any of the notation or its relationship to the study or how it could then be a motivation for the experiment.

2. Two RL models are presented, and applied to the data and their fits are equally good and cannot be differentiated. In addition, the models do not make any different predictions for the experiments either. One is left wondering what the purpose of the exercise is.

3. In the first round, we asked for clarification about which subjects did the 1, 2 or 3-person experiments and how many rounds they did. Here, this request has been addressed but the rebuttal does not really solve any problem. On pp 10-11 (line 172-185) the descriptions are more confusing than helping. For example, in line 184, we are told that 161 individuals participated in triadic experiments but 161 is not divisible by 3. This leaves the reader with the impression that there was no real systematicity to the arrangement of the experimental participation.

I am afraid I cannot be positive

Reviewer #2: The author addressed all of my previous concerns well. Particularly, the modeling approach used and the way the modeling is explained are both much improved.

7. PLOS authors have the option to publish the peer review history of their article (what does this mean?). If published, this will include your full peer review and any attached files.

Reviewer #1: No

Reviewer #2: No

---

## [Author Response · Author response to Decision Letter 1]

25 Apr 2021

We greatly appreciate reviewers’ helpful comments. This time, Reviewer 2 seemed to accept the revision. So current revision was made in response to Reviewer 1’ s comments. 

Main changes

First, the main revisions in the current manuscript are as follows:

-The subsample in the previous manuscript was too strict to severely limit the sample size. So this time, we relaxed the criteria of selecting the subsample to compare across different groups. This time, we selected the sample according to the criteria in which at least one member of the dyadic and triadic groups undertook Test 3 and Test 2, respectively. The results remained the same as before, but this time, the size of this subsample was 262 individuals, 116 dyadic groups and 93 triadic groups, much larger than the previous subsample.

- We described our hypothesis more clearly in Introduction, and moved the algebraic model part to Appendix. 

- In the previous manuscript, we separately examined learning rates of α^±. But this time, we calculated the positivity biases α^+-α^- because many related studies reported positivity biases emerged for many participants. In this manuscript, it was revealed that positivity biases were found in individuals and triadic groups, but not in dyadic groups. This could be one evidence for learning incoherence in dyadic groups. We also examined this positivity biases in within group effects. 

- The description regarding the number of participants who took tests several times seemed to confuse Reviewer 1. So we rewrote the relevant part to clarify the fact that in each group, some participants undertook tests more than once. 

Responses to Reviewer 1:

1. “In the introduction, we see an attempt at motivating a theoretical justification in pp. 6-7. Frankly, I did not understand any of the notation or its relationship to the study or how it could then be a motivation for the experiment.”

- First, we rewrote this part to make it clearer. The hypothesis was as follows:

- Odd numbered group sizes (individuals and triads) generate higher performance due to learning coherence. Even numbered group sizes (dyads) generate lower performance due to learning incoherence. Learning coherence emerged because majority groups could take initiative in decision making over time. In triadic groups, two members out of three could form majority groups and make decision. In individuals, only one member made decision. Learning incoherence was generated due to no such majority subgroups emerged, especially in dyadic groups. In this case, two members agreed or two members did not agree. 

- The simple algebraic model just described this situation and compared expected rewards in individuals, dyads, and triads. 

- In the empirical study, first we compared performance across individuals, dyads, and triads. Then, we compared learning properties. We interpreted learning incoherence took place in dyads because the positivity biases did not emerge and the inverse temperature was somewhere between individuals and triads. In triads, they pursue more exploitation strategy. In individuals, they pursue more exploratory strategy. However, dyadic groups took a stuck-in-the-middle strategy, which we interpret as one evidence of learning incoherence. Thus, we believe our hypothesis was supported in the empirical study: Individuals and triads generated higher performance due to learning coherence and dyads generated lower performance due to learning incoherence. 

- As for the model part, while it was moved to Appendix, we could also drop this part, if reviewers consider it unnecessary. The detailed explanation was provided in the file "Revision List". 

2. “Two RL models are presented, and applied to the data and their fits are equally good and cannot be differentiated. In addition, the models do not make any different predictions for the experiments either. One is left wondering what the purpose of the exercise is.”

- The two RL models were prepared in response to Reviewer 1’s request for the necessity of model comparison. The result was undifferentiated between the two models. We think two responses are possible. One is to drop either model. Since the positivity biases could not be evaluated in the simple model, this model could be dropped. Another possibility is to maintain two models, and examine whether the learning properties remain the same between the two alternative models. We took this alternative and check the robustness of the results. The results indicated that inverse temperatures accounted for higher performance in both models and we believe they show the robustness of the results. Although reporting the results of the two models seemed more persuasive, if Reviewer 1 prefer to drop either model, we will drop the simple model and report the result of the asymmetric model alone. 

3. “In the first round, we asked for clarification about which subjects did the 1, 2 or 3-person experiments and how many rounds they did. Here, this request has been addressed but the rebuttal does not really solve any problem. On pp 10-11 (line 172-185) the descriptions are more confusing than helping. For example, in line 184, we are told that 161 individuals participated in triadic experiments but 161 is not divisible by 3. This leaves the reader with the impression that there was no real systematicity to the arrangement of the experimental participation.”

- In this description, we are afraid that some misunderstanding exists. Please note that in the subsample, some individuals undertook tests more than once. Thus, 161 individuals participated in triadic groups, but some individuals participated in these groups more than once. Therefore, the number of individuals (=161) was not necessarily the multiples of three. 

- However, taking the fact that our description induced misunderstanding, we rewrote this part with more information. Since we changed the subsample this time, we rewrote the corresponding part. 

- Let us examine the relation across the number of individuals and that of dyadic and triadic groups as follows:

Pooled sample

 # of individuals = 322 

 # of dyads=138 (230 and 23 individuals took once and twice)

 # of triads=108 (153, 66, 13 individuals took one, two, and three times)

In dyads, the total # of individuals is 276 (=138*2). 

 230 individuals once

 # of individuals taking twice = 46 (=23*2)

 Sum: 230+46=276

In triads, the total # of individuals is 324 (=108*3)

 153 individuals once

 # of individuals taking twice = 132 (=66*2)

 # of individuals taking three times = 39 (=13*3)

 Sum: 153+132+39=324

Note that some participants undertook both Tests 2 and 3. We described this information regarding the subsample because it was relevant to group comparison, which used the subsample alone in this manuscript.

In the subsample, 

 # of individuals = 262

 # of dyads=116 (200 and 16 individuals took once and twice)

 # of triads=93 (142, 52, 11 individuals took one, two, and three times)

In dyads, the total # of individuals is 232 (=116*2). 

 200 individuals taking once

 # of individuals taking twice = 32 (=16*2)

 Sum: 200+32=232

In triads, the total # of individuals is 279 (=93*3)

 142 individuals taking once

 # of individuals taking twice = 104 (=52*2)

 # of individuals taking three times = 33 (=11*3)

 Sum: 142+104+33=279

We also provided the information for those taking either or both Tests 2 and 3 in the subsample because this was related to group comparison. 

The total # of individuals taking Tests 2 and 3=511(=116 dyads*2+93 triads*3)

 # of individuals taking these tests once = 84

 # of individuals taking these tests twice=118

 # of individuals taking these test three times = 49

 # of individuals taking these test four times=11

 Sum: Total # of individuals = 84+118*2+49*3+11*4=511

 # of individuals = 84+118+49+11=262

The related information was provided in the following sentences:

“All participants in this sample undertook Test 1 once and at least one more experiment in Tests 2 or 3. In Test 2 (dyads), 230 and 23 participants played the TAB once and twice, respectively. In Test 3 (triads), 153, 66 and 13 participants played one, two, and three times, respectively.”

” In total, 262 individuals were in this subsample. In this subsample, 116 dyadic groups and 93 triadic groups were identified. In Tests 2 and 3, the numbers of individuals who took the tests one, two, three, and four times were 84, 118, 49, and 11, respectively. Of the dyads, 216 individuals participated, and 200 and 16 undertook Test 2 once and twice, respectively. Of the triads, 205 individuals participated, and 142, 52 and 11 individuals undertook Test 3 one, two and three times, respectively.”

---

## [Editor Report · Decision Letter 2]

11 May 2021

Three heads are better than two: Comparing learning properties and performances across individuals, dyads, and triads through a computational approach

PONE-D-20-35580R2

Dear Dr. Harada,

We’re pleased to inform you that your manuscript has been judged scientifically suitable for publication and will be formally accepted for publication once it meets all outstanding technical requirements.

Kind regards,

Darrell A. Worthy, Ph.D

Academic Editor

PLOS ONE
---

## [Editor Report · Acceptance letter]

31 May 2021

PONE-D-20-35580R2 

Three heads are better than two: Comparing learning properties and performances across individuals, dyads, and triads through a computational approach 

Dear Dr. Harada:

I'm pleased to inform you that your manuscript has been deemed suitable for publication in PLOS ONE. Congratulations! Your manuscript is now with our production department. 

Kind regards, 

on behalf of

Dr. Darrell A. Worthy 

Academic Editor

PLOS ONE